# SYNTHESIZING AUDIO WITH GANS

**Chris Donahue\*, Julian McAuley**[†] **& Miller Puckette\***
Departments of Music\* and Computer Science[†]
University of California, San Diego
{cdonahue,jmcauley,msp}@ucsd.edu

## ABSTRACT

While Generative Adversarial Networks (GANs) have seen wide success at the problem of synthesizing realistic images, they have seen little application to audio generation. In this paper, we introduce WaveGAN, a first attempt at applying GANs to raw audio synthesis in an unsupervised setting. Our experiments on speech demonstrate that WaveGAN can produce intelligible words from a small vocabulary of human speech, as well as synthesize audio from other domains such as bird vocalizations, drums, and piano. Qualitatively, we find that human judges prefer the generated examples from WaveGAN over those from a method which naïvely applies GANs on image-like audio feature representations.

## 1 INTRODUCTION

Synthesizing audio for specific domains has many practical applications such as text-to-speech and music production. End-to-end, learning-based approaches have recently eclipsed the performance of production parametric systems in the area of text-to-speech (Wang et al., 2017). Such methods depend on access to large quantities of transcribed recordings, but do not take advantage of additional untranscribed audio that is often available. Unsupervised approaches may be able to reduce data requirements for these methods by learning to synthesize *a priori*. However, audio signals have high temporal resolution with periodic behavior over large windows, and relevant unsupervised strategies must perform effectively in high dimensions. Recent work has demonstrated that convolutional (Oord et al., 2016) and recurrent (Mehri et al., 2017) neural networks can be trained with autoregression to learn to generate raw audio. However, generation with these methods is slow as the network must be evaluated once per audio sample.

Unlike autoregressive approaches, generative adversarial networks (Goodfellow et al., 2014) can generate high-dimensional signals more quickly by iteratively upsampling low-dimensional noise vectors with transposed convolution (Radford et al., 2016). Since their introduction, GANs have been refined to generate images with increasing fidelity (Berthelot et al., 2017; Karras et al., 2018). Despite their prevalence for image applications, GANs have yet to be demonstrated capable of synthesizing audio in an unsupervised setting.

In this work, we propose WaveGAN, a time-domain strategy for generating slices of raw audio with GANs.[1] WaveGAN adapts the DCGAN architecture (Radford et al., 2016), which popularized GANs for image synthesis, for operation on raw audio. Our modifications may serve as a template for adapting other image-processing architectures to audio. We also propose SpecGAN, an approach to generating audio in the frequency domain, which naïvely trains the DCGAN architecture on image-like audio *spectrograms*.

To evaluate our methods, we propose a new standard task, generating spoken examples of digits "zero" through "nine." We design our evaluation methodology based on scores from a pre-trained classifier and human judgements. Our experiments on this task demonstrate that both WaveGAN and SpecGAN can generate examples of speech that are intelligible to humans. On criteria of sound quality and speaker diversity, human judges indicate a preference for the examples from WaveGAN compared to those from SpecGAN.

---

[1]Sound examples (https://goo.gl/7EH4Z8). IPython notebook (https://goo.gl/ChSPp9).

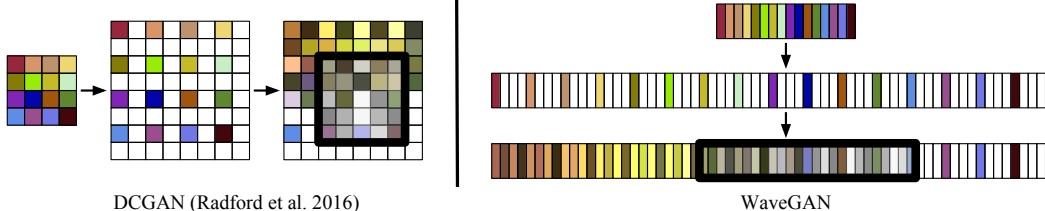

Figure 1: Depiction of the transposed convolution operation for the first layers of the DCGAN (Radford et al., 2016) (**left**) and WaveGAN (**right**) generators. DCGAN uses small (5x5), two-dimensional filters while WaveGAN uses longer (length-25), one-dimensional filters and a larger upsampling factor. Both have the same number of parameters and numerical operations.

## 2 WAVEGAN

We base our WaveGAN approach on DCGAN (Radford et al., 2016), modifying the architecture to operate in one dimension. Specifically, we use longer one-dimensional filters of length 25 instead of two-dimensional filters of size 5x5, and we upsample by a factor of 4 instead of 2 at each layer (Figure 1). We modify the discriminator in a similar way, using length-25 filters in one dimension and increasing stride from 2 to 4. Out of several GAN-training algorithms, we found that the WGAN-GP (Gulrajani et al., 2017) strategy alone trained WaveGAN to produce reasonable outputs.

Because DCGAN outputs 64x64 pixel images — equivalent to just 4096 audio samples — we add one additional layer to the model resulting in 16384 samples, slightly more than one second of audio at 16 kHz. While 16k samples is a sufficient length for certain sound domains (e.g. sound effects, voice commands), generalization to longer output is an avenue for future work. A complete description of our model and training hyper parameters can be found in the appendix.

### 2.1 PHASE SHUFFLE

Transposed convolution is known to produce characteristic "checkerboard artifacts" in images (Odena et al., 2016). For audio, analogous artifacts are perceived as tones with particular frequencies. These artifacts have consistent phase in the generated examples, and the discriminator could learn a trivial solution that rejects them on this basis, inhibiting the overall optimization problem. To discourage the discriminator from learning such a solution, we propose the *phase shuffle* operation (with hyperparameter $n$) which randomly perturbs the phase of each layer's activations by $-n$ to $n$ samples before input to the next layer. See (Donahue et al., 2018) for a complete description.

## 3 SPECGAN

For SpecGAN, our frequency-domain audio generation model, we design a spectrogram representation that is both well-suited to GANs designed for image generation and *can* be approximately inverted. Our representation is a log-amplitude magnitude spectrum of the short-time Fourier transform, normalized to $[-1, 1]$. We choose a window size of 16 ms with 8 ms stride, resulting in equivalent dimensionality to the time domain (16384 samples yield a 128x128 spectrogram). To render the resultant generated spectrograms as waveforms, we employ the iterative Griffin-Lim algorithm (Griffin & Lim, 1984) with 16 iterations.

## 4 EXPERIMENTS

Our primary experimentation focuses the *Speech Commands Dataset* (Warden, 2017). This dataset consists of many speakers recording individual words in uncontrolled recording conditions. We propose the *Speech Commands Zero Through Nine* (SC09) subset, which reduces the vocabulary of the dataset to ten words: the digits "zero" through "nine". We also experiment on four other datasets (Figure 2) with different characteristics: 1) large vocabulary speech (TIMIT (Garofolo et al., 1993)), 2) bird vocalizations (Boesman, 2018), 3) single drum hits, and 4) polyphonic piano.

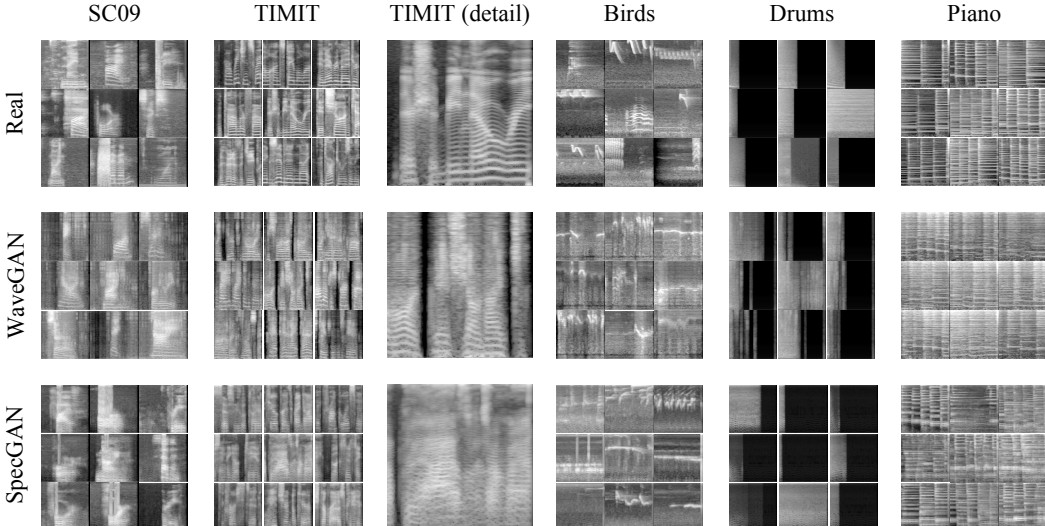

Figure 2: **Top**: Random samples from each of the five datasets used in this study, illustrating the wide variety of spectral characteristics. **Middle**: Random samples generated by WaveGAN for each domain. WaveGAN operates in the time domain but results are displayed here in the frequency domain for visual comparison. **Bottom**: Random samples generated by SpecGAN for each domain.

Table 1: Results for SC09 experiments comparing real and generated data. A higher inception score suggests that semantic modes (i.e., words) of the real data distribution have been captured.

| Dataset | Inception score | Human accuracy | $MOS_Q$ | $MOS_D$ |
|---|---|---|---|---|
| Real data | $8.01 \pm 0.24$ | .976 | 3.95 | 3.54 |
| WaveGAN | $4.12 \pm 0.03$ | | | |
| + Phase shuffle $n = 2$ | $4.67 \pm 0.01$ | .943 | 2.29 | 3.24 |
| SpecGAN | $6.03 \pm 0.04$ | .945 | 1.87 | 2.64 |

## 4.1 EVALUATION METHODOLOGY

We evaluate our SC09 generative models using *inception score* (Szegedy et al., 2016), which measures both the diversity and semantic discriminability of generated examples. To compute inception score, we pre-train a classifier that achieves 93% accuracy on the SC09 test set. We also measure the ability of human annotators on *Amazon Mechanical Turk* to label the generated audio. For data generated by WaveGAN and SpecGAN, we use the classifier's prediction as ground truth.

## 5 RESULTS AND DISCUSSION

Results for our evaluation appear in Table 1. WaveGAN trained with phase shuffle achieved a better inception score than WaveGAN without. Despite the higher inception score for SpecGAN compared to WaveGAN, human judges are able to label examples from the two methods with identical accuracy. Furthermore, on subjective criteria of sound quality ($MOS_Q$) and speaker diversity ($MOS_D$), human annotators assign a higher mean opinion score (MOS) to examples from WaveGAN compared to those from SpecGAN.

In Figure 2, we show results for all our experimental domains. For TIMIT, a large-vocabulary speech dataset with many speakers, WaveGAN produces speech-like babbling (similar to results from unconditional autoregressive models (Oord et al., 2016)). On bird vocalizations, WaveGAN generates a variety of bird sounds but with more noise than the other domains. For drum sound effects, WaveGAN captures semantic modes such as kick and snare drums. On piano, WaveGAN produces musically-consonant motifs that, as with the training data, represent a variety of key signatures.

ACKNOWLEDGMENTS

The authors thank Peter Boesman, Sander Dieleman, and Colin Raffel for helpful conversations about this work. This work was supported by the UC San Diego Department of Computer Science. GPUs used for this work were provided by the HPC @ UC program and donations from NVIDIA.

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

Table 2: WaveGAN generator architecture

| Operation | Kernel Size | Output Shape |
|---|---|---|
| Input $z \sim \text{Uniform}(-1, 1)$ | | $(n, 100)$ |
| Dense 1 | $(100, 256d)$ | $(n, 256d)$ |
| Reshape | | $(n, 16, 16d)$ |
| ReLU | | $(n, 16, 16d)$ |
| Trans Conv1D (Stride=4) | $(25, 16d, 8d)$ | $(n, 64, 8d)$ |
| ReLU | | $(n, 64, 8d)$ |
| Trans Conv1D (Stride=4) | $(25, 8d, 4d)$ | $(n, 256, 4d)$ |
| ReLU | | $(n, 256, 4d)$ |
| Trans Conv1D (Stride=4) | $(25, 4d, 2d)$ | $(n, 1024, 2d)$ |
| ReLU | | $(n, 1024, 2d)$ |
| Trans Conv1D (Stride=4) | $(25, 2d, d)$ | $(n, 4096, d)$ |
| ReLU | | $(n, 4096, d)$ |
| Trans Conv1D (Stride=4) | $(25, d, c)$ | $(n, 16384, c)$ |
| Tanh | | $(n, 16384, c)$ |

Table 3: WaveGAN discriminator architecture

| Operation | Kernel Size | Output Shape |
|---|---|---|
| Input $x$ or $G(z)$ | | $(n, 16384, c)$ |
| Conv1D (Stride=4) | $(25, c, d)$ | $(n, 4096, d)$ |
| LReLU ($\alpha = 0.2$) | | $(n, 4096, d)$ |
| Phase Shuffle ($n = 2$) | | $(n, 4096, d)$ |
| Conv1D (Stride=4) | $(25, d, 2d)$ | $(n, 1024, 2d)$ |
| LReLU ($\alpha = 0.2$) | | $(n, 1024, 2d)$ |
| Phase Shuffle ($n = 2$) | | $(n, 1024, 2d)$ |
| Conv1D (Stride=4) | $(25, 2d, 4d)$ | $(n, 256, 4d)$ |
| LReLU ($\alpha = 0.2$) | | $(n, 256, 4d)$ |
| Phase Shuffle ($n = 2$) | | $(n, 256, 4d)$ |
| Conv1D (Stride=4) | $(25, 4d, 8d)$ | $(n, 64, 8d)$ |
| LReLU ($\alpha = 0.2$) | | $(n, 64, 8d)$ |
| Phase Shuffle ($n = 2$) | | $(n, 64, 8d)$ |
| Conv1D (Stride=4) | $(25, 8d, 16d)$ | $(n, 16, 16d)$ |
| LReLU ($\alpha = 0.2$) | | $(n, 16, 16d)$ |
| Reshape | | $(n, 256d)$ |
| Dense | $(256d, 1)$ | $(n, 1)$ |

## A  ARCHITECTURE DESCRIPTION

In Tables 2 and 3, we list the full architectures for our WaveGAN generator and discriminator respectively. In Tables 4 and 5, we list the full architectures for our SpecGAN generator and discriminator respectively. In these tables, $n$ is the batch size, $d$ modifies model size, and $c$ is the number of channels in the examples. All dense and convolutional layers include biases.

## B  TRAINING HYPERPARAMETERS

In Table 6, we list the values of these and all other hyperparameters for our experiments, which constitute our out-of-the-box recommendations for applying WaveGAN and SpecGAN to new datasets.

Table 4: SpecGAN generator architecture

| Operation | Kernel Size | Output Shape |
|---|---|---|
| Input $z \sim \text{Uniform}(-1, 1)$ | | $(n, 100)$ |
| Dense 1 | $(100, 256d)$ | $(n, 256d)$ |
| Reshape | | $(n, 4, 4, 16d)$ |
| ReLU | | $(n, 4, 4, 16d)$ |
| Trans Conv1D (Stride=2) | $(5, 5, 16d, 8d)$ | $(n, 8, 8, 8d)$ |
| ReLU | | $(n, 8, 8, 8d)$ |
| Trans Conv1D (Stride=2) | $(5, 5, 8d, 4d)$ | $(n, 16, 16, 4d)$ |
| ReLU | | $(n, 16, 16, 4d)$ |
| Trans Conv1D (Stride=2) | $(5, 5, 4d, 2d)$ | $(n, 32, 32, 2d)$ |
| ReLU | | $(n, 32, 32, 2d)$ |
| Trans Conv1D (Stride=2) | $(5, 5, 2d, d)$ | $(n, 64, 64, d)$ |
| ReLU | | $(n, 64, 64, d)$ |
| Trans Conv1D (Stride=2) | $(5, 5, d, c)$ | $(n, 128, 128, c)$ |
| Tanh | | $(n, 128, 128, c)$ |

Table 5: SpecGAN discriminator architecture

| Operation | Kernel Size | Output Shape |
|---|---|---|
| Input $x$ or $G(z)$ | | $(n, 128, 128, c)$ |
| Conv1D (Stride=4) | $(5, 5, c, d)$ | $(n, 64, 64, d)$ |
| LReLU ($\alpha = 0.2$) | | $(n, 64, 64, d)$ |
| Conv1D (Stride=4) | $(5, 5, d, 2d)$ | $(n, 32, 32, 2d)$ |
| LReLU ($\alpha = 0.2$) | | $(n, 32, 32, 2d)$ |
| Conv1D (Stride=4) | $(5, 5, 2d, 4d)$ | $(n, 16, 16, 4d)$ |
| LReLU ($\alpha = 0.2$) | | $(n, 16, 16, 4d)$ |
| Conv1D (Stride=4) | $(5, 5, 4d, 8d)$ | $(n, 8, 8, 8d)$ |
| LReLU ($\alpha = 0.2$) | | $(n, 8, 8, 8d)$ |
| Conv1D (Stride=4) | $(5, 5, 8d, 16d)$ | $(n, 4, 4, 16d)$ |
| LReLU ($\alpha = 0.2$) | | $(n, 4, 4, 16d)$ |
| Reshape | | $(n, 256d)$ |
| Dense | $(256d, 1)$ | $(n, 1)$ |

Table 6: WaveGAN hyperparameters

| Name | Value |
|---|---|
| Input data type | 16-bit PCM (requantized) |
| Model data type | 32-bit floating point |
| Num channels ($c$) | 1 |
| Batch size ($b$) | 64 |
| Model size ($d$) | 64 |
| Phase shuffle (WaveGAN) | 2 |
| Phase shuffle (SpecGAN) | 0 |
| Loss | WGAN-GP (Gulrajani et al., 2017) |
| WGAN-GP $\lambda$ | 10 |
| $D$ updates per $G$ update | 5 |
| Optimizer | Adam (Kingma & Ba, 2015) |
| Learning rate | 0.0001 |
| Beta 1 | 0.5 |
| Beta 2 | 0.9 |

