# OpenReview forum: "Synthesizing Audio with GANs"
_ICLR.cc/2018/Workshop — Accept_

### Official Review · AnonReviewer2 · 2018-03-08
**Good application of GAN on audio data; more experiments encouraged.**

**Rating:** 6
**Confidence:** 4

**Review:**

Summary:
- This paper applies GAN (specifically DCGAN) on the task of audio generation. Two variants of model are considered, which generate on time domain and frequency domain. The author show that frequency domain generation leads to better semantics (better inception score) but lower sound quality.

Strength:
- WaveGAN and SpecGAN are two sensible formulation of audio synthesis models.

- Phase shuffle is a neat trick for avoiding too powerful discriminator.

Weakness:
- The story of the paper is not clear. The abstract is all about WaveGAN; however, It seems that the SpecGAN leads to better performance on inception score, which is also not given enough discussion: e.g. why is SpecGAN has better inception score but lower MOS?

- Only quantitative results for SC09 is provided, which is a pretty small dataset.

- No comparison to other audio synthesizer approaches.

Conclusion:
- The authors have shown promising experimental results for two GAN based models for audio synthesis. More experimental results can make this paper more solid. As of the current form, I recommend weak accept.

---

### Official Review · AnonReviewer1 · 2018-03-09
**The paper presents a novel application of Generative Adversarial Networks for raw audio synthesis in an unsupervised setting.**

**Rating:** 7
**Confidence:** 4

**Review:**

The paper discusses Generative Adversarial Networks for synthesizing raw audio slices. It proposes a new task for evaluating this approach, that is, the generation of spoken digits from zero to nine. The work presents two variants of GAN for this problem - WaveGAN, which adapts DC-GAN for 1 dimensional input, and SpecGAN, which takes audio spectrograms as image-like 2D input. Quantitatively, the audio slices are good ( fair inception score and human accuracy ), but are poor qualitatively (low MOS). The work establishes that GANs can learn semantically meaningful modes for small vocabulary speech sets.

Paper is well written. The work is new and significant in that there are no precedents of GANs applied to raw audio synthesis. The work proposes a new architecture and evaluation methodology for this problem, and successfully demonstrates the feasibility of GANs in this domain. This is an exciting new direction that should be explored further.

---

### Official Review · AnonReviewer3 · 2018-03-14
**Early results of audio synthesis with GANs**

**Rating:** 7
**Confidence:** 5

**Review:**

This paper presents early results for synthesizing audio with GANs, The paper also contributes of a new speech synthesis dataset called SC09 comprising of samples of people speaking 0 through 9, which is used to evaluate the proposed model and could also be useful to debug and evaluate speech synthesis models in general. Apart from SC09, the evaluations use TIMIT, bird vocalizations, single drum hits, and polyphonic music.

The simplicity of the SC09 dataset allows the authors to quantitatively measure the performance of the models using the Inception score. Inception scores for WaveGAN and SpecGAN are given, however there is no comparison with other baseline synthesis models such as Wavenet, Tacotron, SampleRNN, etc.

The authors provide audio samples and an ipython notebook comprising of inference code and model architecture and weights. This encourages other researchers to build on top of this work and/or compare, and is in good spirit of a workshop submission.

---

### Decision · Program_Chairs · 2018-03-20
**ICLR 2018 Workshop Acceptance Decision**

**Decision:**

Accept

**Comment:**

Congratulations, your paper was accepted to the ICLR workshop.